# REDUCING TRANSFORMER DEPTH ON DEMAND WITH STRUCTURED DROPOUT

**Angela Fan**
Facebook AI Research/LORIA
angelafan@fb.com

**Edouard Grave**
Facebook AI Research
egrave@fb.com

**Armand Joulin**
Facebook AI Research
ajoulin@fb.com

## ABSTRACT

Overparameterized transformer networks have obtained state of the art results in various natural language processing tasks, such as machine translation, language modeling, and question answering. These models contain hundreds of millions of parameters, necessitating a large amount of computation and making them prone to overfitting. In this work, we explore *LayerDrop*, a form of structured dropout, which has a regularization effect during training and allows for efficient pruning at inference time. In particular, we show that it is possible to select sub-networks of any depth from one large network without having to finetune them and with limited impact on performance. We demonstrate the effectiveness of our approach by improving the state of the art on machine translation, language modeling, summarization, question answering, and language understanding benchmarks. Moreover, we show that our approach leads to small BERT-like models of higher quality compared to training from scratch or using distillation.

## 1 INTRODUCTION

Transformer architectures (Vaswani et al., 2017) have become the dominant architecture in natural language processing, with state-of-the-art performance across a variety of tasks, including machine translation (Vaswani et al., 2017; Ott et al., 2018), language modeling (Dai et al., 2019; Baevski & Auli, 2018) and sentence representation (Devlin et al., 2018; Yang et al., 2019). Each of its layers contains millions of parameters accessed during the forward pass, making it computationally demanding in terms of memory and latency during both training and inference. In an ideal situation, we would be able to extract sub-networks — automatically and without finetuning — from this over-parameterized network, for any given memory or latency constraint, while maintaining good performance. In contrast, standard pruning or distillation methods follow a strategy that often includes a finetuning or retraining step, and the process must be repeated for each desired depth.

In this work, we propose a novel approach to extract *any* sub-network without a post-hoc pruning process from over-parameterized networks. The core of our method is to sample small sub-networks from the larger model during training by randomly dropping model weights as in Dropout (Hinton et al., 2012) or DropConnect (Wan et al., 2013). This has the advantage of making the network robust to subsequent pruning. If well-chosen groups of weights are dropped simultaneously, the resulting small sub-networks can be very efficient. In particular, we drop entire layers to extract shallow models at inference time. Previous work (Huang et al., 2016) has shown that dropping layers during training can regularize and reduce the training time of very deep convolutional networks. In contrast, we focus on pruning. As illustrated in Figure 1, an advantage of our layer dropping technique, or *LayerDrop*, is that from one single deep model, we can extract shallow sub-networks of any desired depth on demand at inference time.

We validate our findings on a variety of competitive benchmarks, namely WMT14 English-German for machine translation, WikiText-103 (Merity et al., 2016) for language modeling, CNN-Dailymail (Hermann et al., 2015) for abstractive summarization, ELI5 (Fan et al., 2017) for long form question answering, and several natural language understanding tasks (Wang et al., 2019a) for sentence representation. Our approach achieves state of the art on most of these benchmarks as a result of the regularization effect, which stabilizes the training of larger and deeper networks. We also show that we can prune Transformer architectures to much smaller models while maintaining com-

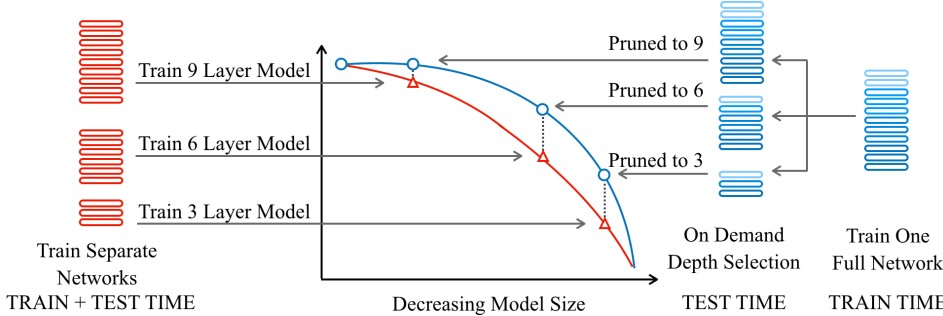

Figure 1: **LayerDrop** (right) randomly drops layers at training time. At test time, this allows for sub-network selection to any desired depth as the network has been trained to be robust to pruning. In contrast to standard approaches that must re-train a new model from scratch for each model size (left), our method trains only one network from which multiple shallow models can be extracted.

petitive performance, outperforming specific model reduction strategies dedicated to BERT (Devlin et al., 2018; Sanh, 2019) as well as training smaller models from scratch. Overall, applying Layer-Drop to Transformer networks provides the following key advantages:

- LayerDrop regularizes very deep Transformers and stabilizes their training, leading to state-of-the-art performance across a variety of benchmarks.

- Small and efficient models of any depth can be extracted automatically at test time from a single large pre-trained model, without the need for finetuning.

- LayerDrop is as simple to implement as dropout.

## 2 RELATED WORK

Our approach is a form of Dropout (Srivastava et al., 2014) applied to model weights instead of activations, as in DropConnect (Wan et al., 2013). Different from DropConnect, we drop groups of weights to induce group redundancy to create models suited for pruning to shallow, efficient models at inference time. Gomez et al. (2018) propose a targeted Dropout and DropConnect, where they learn the drop rate of the weights to match a targeted pruning scheme. Instead, we adapt the masks to the structures that we are interested in pruning. Closer to our work, the Stochastic Depth approach of Huang et al. (2016) drops layers randomly during training. As opposed to our work, they are interested in accelerating the training of very deep ResNets (He et al., 2016), so their dropping schedule is adapted to this goal. Concurrently to this work, Pham et al. (2019) applied Stochastic Depth to train very deep Transformers for speech and show the benefits of its regularization effect.

More generally, our method is a form of structured pruning (Liu et al., 2018b). As opposed to weight pruning (LeCun et al., 1990), structured pruning removes coherent groups of weights to preserve the original structure of the network. Structured pruning has been used in some NLP applications, such as machine translation (See et al., 2016), text classification (Joulin et al., 2016) and language modeling (Murray & Chiang, 2015). However, it has been more widely adopted in computer vision and applied to convolutional network to remove filters (Li et al., 2016; Wen et al., 2016), channels (He et al., 2017), or residual blocks (Huang et al., 2018; Huang & Wang, 2018). Similar to Mittal et al. (2018), we take advantage of the plasticity of neural networks to learn models that are resilient to random pruning or skipping connections Wang et al. (2018); Wu et al. (2018); Liu et al. (2018a), rather than learning the pruning itself. We refer the reader to Liu et al. (2018b) for an exhaustive study of these approaches and their evaluation in the context of convolutional networks.

Reducing the memory footprint of Transformer architectures and BERT in particular is an active subject of research. Several works have compressed BERT as a post-processing step using different forms of distillation (Turc et al., 2019; Tang et al., 2019; Shulga, 2019; Sanh, 2019). Similarly, various papers have shown evidence that Transformers are over-parameterized, especially that most self-attention heads can be dropped at test time (Michel et al., 2019; Voita et al., 2019). Different

from these, our models are trained to be resilient to pruning, which significantly reduces the performance drop induced by test time pruning. Others have proposed trainable adaptive mechanisms to control their memory footprint (Jernite et al., 2016; Sukhbaatar et al., 2019; Correia et al., 2019). These approaches are complementary to ours and should benefit from each other.

## 3 METHOD

In this section, we briefly introduce the Transformer, then describe our Structured Dropout technique and its application to layers. We also discuss several inference time pruning strategies.

### 3.1 THE TRANSFORMER ARCHITECTURE

We succinctly review the Transformer architecture and refer the reader to Vaswani et al. (2017) for additional details. A Transformer is a stack of layers composed of two sub-layers: multi-head self-attention followed by a feedforward sub-layer. The multi-head self-attention sub-layer consists of multiple attention heads applied in parallel. Each attention head takes a matrix $\mathbf{X}$ where each row represents an element of the input sequence and updates their representations by gathering information from their context using an Attention mechanism (Bahdanau et al., 2014):

$$\mathbf{Y} = \text{Softmax}(\mathbf{X}^T\mathbf{K}(\mathbf{Q}\mathbf{X} + \mathbf{P}))\mathbf{V}\mathbf{X},$$

where $\mathbf{K}$, $\mathbf{V}$, $\mathbf{Q}$ and $\mathbf{P}$ are matrices of parameters. The outputs of the heads are then concatenated along the time step into a sequence of vectors.

The second sub-layer then applies a fully connected feedforward network to each element of this sequence independently, $\text{FFN}(\mathbf{x}) = \mathbf{U}\,\texttt{ReLU}\,(\mathbf{V}\mathbf{x})$, where $\mathbf{V}$ and $\mathbf{U}$ are matrices of parameters. Each sub-layer is followed by a `AddNorm` operation that is a residual connection (He et al., 2016) and a layer normalization (Ba et al., 2016).

### 3.2 TRAINING TRANSFORMERS WITH RANDOM STRUCTURED PRUNING

We present a regularization approach that makes Transformers robust to subsequent structured pruning at inference time. We focus in particular on the case where the targeted structure is a layer.

#### 3.2.1 RANDOMLY DROPPING STRUCTURES AT TRAINING TIME

Regularizing networks to be robust to pruning can be achieved by randomly removing weights during its training as in DropConnect (Wan et al., 2013). In this approach, each weight is dropped independently following a Bernoulli distribution associated with a parameter $p > 0$ that controls the drop rate. This is equivalent to a pointwise multiplication of the weight matrix $\mathbf{W}$ with a randomly sampled $\{0, 1\}$ mask matrix $\mathbf{M}$:

$$\mathbf{W}_d = \mathbf{M} \odot \mathbf{W}.$$

DropConnect is a form of random unstructured pruning that leads to smaller, but not necessarily more efficient, models. We propose to add structure to this mechanism to target model efficiency.

**Random Structured Dropout.** The weights of a Transformer network belong to multiple overlapping structures, such as heads, FFN matrices, or layers. Dropping weights using groups that follow some of these inherent structures potentially leads to a significant reduction of the inference time. This is equivalent to constraining the mask $\mathbf{M}$ to be constant over some predefined groups of weights. More precisely, given a set $\mathcal{G}$ of predefined groups of weights, the $\{0, 1\}$ mask matrix $\mathbf{M}$ is randomly sampled over groups instead of weights:

$$\forall i,\ \mathbf{M}[i] \in \{0, 1\}, \quad \text{and}\ \forall G \in \mathcal{G},\ \forall (i, j) \in G,\ \mathbf{M}[i] = \mathbf{M}[j].$$

This structured dropout formulation is general and can be applied to any overlapping groups of weights, whether heads, FFN matrices, or layers. Nonetheless, not all of the structures in a Transformer lead to the same benefits when dropped. For example, dropping attention heads does not reduce runtime as they are usually computed in parallel. For simplicity, we focus on dropping layers, and we name this structured pruning, *LayerDrop*. This is inspired by the Stochastic Depth approach of Huang et al. (2016) used to train very deep ResNets (He et al., 2015).

### 3.2.2 PRUNING AT INFERENCE TIME

**Selecting Layers to Prune**    Training with LayerDrop makes the network more robust to predicting with missing layers. However, LayerDrop does not explicitly provide a way to select which groups to prune. We consider several different pruning strategies, described below:

- *Every Other*: A straightforward strategy is to simply drop every other layer. Pruning with a rate $p$ means dropping the layers at a depth $d$ such that $d \equiv 0(\mathrm{mod}\lfloor\frac{1}{p}\rfloor)$. This strategy is intuitive and leads to balanced networks.

- *Search on Valid*: Another possibility is to compute various combinations of layers to form shallower networks using the validation set, then select the best performing for test. This is straightforward but computationally intensive and can lead to overfitting on validation.

- *Data Driven Pruning*: Finally, we propose *data driven pruning* where we learn the drop rate of each layer. Given a target drop rate $p$, we learn an individual drop rate $p_d$ for the layer at depth $d$ such that the average rate over layers is equal to $p$. More precisely, we parameterize $p_d$ as a non-linear function of the activation of its layer and apply a softmax. At inference time, we forward only the fixed top-k highest scoring layers based on the softmax output (e.g. chosen layers do not depend on the input features).

In practice, we observe that the *Every Other* strategy works surprisingly well across many tasks and configurations. *Search on Valid* and *Data Driven Pruning* only offer marginal gains. Note that we do not further finetune any of the pruned networks (see Appendix for analysis of finetuning).

**Setting the drop rate for optimal pruning.**    There is a straightforward relationship between the drop rate of groups and the average pruning level that the network should be resilient to. Assuming $N$ groups and a fixed drop ratio $p$, the average number of groups used by the network during training is $N(1 - p)$. As a consequence, to target a pruning size of $r$ groups, the optimal drop rate is:

$$p^* = 1 - \frac{r}{N}$$

In practice, we observe that networks are more robust to pruning than their expected ratio but higher pruning rates leads to better performance for smaller models. We use a LayerDrop rate of $p = 0.2$ for all our experiments, but we recommend $p = 0.5$ to target very small inference time models.

## 4   EXPERIMENTAL SETUP

We apply our method to a variety of sequence modeling tasks: neural machine translation, language modeling, summarization, long form question answering, and various natural language understanding tasks. Our models are implemented in PyTorch using `fairseq-py` (Ott et al., 2019)[1]. Additional implementation and training details with hyperparameter settings are in the Appendix.

**Neural Machine Translation.**    We experiment on the WMT English-German machine translation benchmark using the Transformer Big architecture. We use the dataset of $4.5$M en-de sentence pairs from WMT16 (Vaswani et al., 2017) for training, newstest2013 for validation, and newstest2014 for test. We optimize the dropout value within the range $\{0.1, 0.2, 0.5\}$ on the validation set and set the LayerDrop rate $p$ to $0.2$. For generation, we average the last $10$ checkpoints, set the length penalty to $0.6$, and beam size to $8$, following the settings suggested in Wu et al. (2019a), and measure case-sensitive tokenized BLEU. We apply compound splitting, as used in Vaswani et al. (2017).

**Language Modeling.**    We experiment on the Wikitext-103 language modeling benchmark (Merity et al., 2016) which contains 100M tokens and a large vocabulary size of 260K. We adopt the 16 layer Transformer used in Baevski & Auli (2018). We set the LayerDrop rate $p$ to $0.2$ and tune the standard dropout parameter in $\{0.1, 0.2, 0.3\}$ on the validation set. We report test set perplexity (PPL).

---

[1]`https://github.com/pytorch/fairseq/tree/master/examples/layerdrop`

| Model | Enc Layers | Dec Layers | BLEU |
|---|---|---|---|
| Transformer (Vaswani et al., 2017) | 6 | 6 | 28.4 |
| Transformer (Ott et al., 2018) | 6 | 6 | 29.3 |
| DynamicConv (Wu et al., 2019a) | 7 | 6 | 29.7 |
| Transformer (Ott et al., 2018) + LayerDrop | 6 | 6 | 29.6 |
| Transformer (Ott et al., 2018) + LayerDrop | 12 | 6 | **30.2** |

Table 1: **Results on WMT en-de Machine Translation** (newstest2014 test set)

| Model | Layers | Params | PPL |
|---|---|---|---|
| Adaptive Inputs (Baevski & Auli, 2018) | 16 | 247M | 18.7 |
| Transformer XL Large (Dai et al., 2019) | 18 | 257M | 18.3 |
| Adaptive Inputs + LayerDrop | 16 | 247M | 18.3 |
| Adaptive Inputs + LayerDrop | 40 | 423M | **17.7** |

Table 2: **Results on Wikitext-103** language modeling benchmark (test set).

**Summarization.**   We adopt the Transformer base architecture and training schedule from Edunov et al. (2019) and experiment on the CNN-Dailymail multi-sentence summarization benchmark. The training data contains over 280K full-text news articles paired with multi-sentence summaries (Hermann et al., 2015; See et al., 2017). We tune a generation length in the range $\{40, 50, 60\}$ and use 3-gram blocking. We set the LayerDrop rate $p$ to $0.2$. We evaluate using ROUGE (Lin, 2004).

**Long Form Question Answering.**   We consider the Long Form Question Answering Dataset ELI5 of Fan et al. (2019), which consists of 272K question answer pairs from the subreddit *Explain Like I'm Five* along with extracted supporting documents from web search. We follow the Transformer Big architecture and training procedure of Fan et al. (2019). We generate long answers using beam search with beam size $5$ and apply 3-gram blocking (Fan et al., 2017). We evaluate with ROUGE.

**Sentence representation Pre-training.**   We train base and large BERT (Devlin et al., 2018) models following the open-source implementation of Liu et al. (2019). We use two datasets: Bookscorpus + Wiki from Liu et al. (2019) and the larger combination of Bookscorpus + OpenWebText + CC-News + Stories (Liu et al., 2019). We evaluate the pretrained models on various natural language understanding tasks. Specifically, we evaluate accuracy on MRPC (Dolan & Brockett, 2005), QNLI (Rajpurkar et al., 2016), MNLI (Williams et al., 2018), and SST2 (Socher et al., 2013).

## 5 RESULTS

### 5.1 LAYERDROP AS A REGULARIZER

**Language Modeling.**   In Table 2, we show the impact of LayerDrop on the performance of a Transformer network trained in the setting of Adaptive Inputs (Baevski & Auli, 2018). Adding LayerDrop to a 16 layer Transformer improves the performance by $0.4$ perplexity, matching the state-of-the-art results of Transformer-XL. Our 40 layer Transformer with LayerDrop further improves the state of the art by $0.6$ points. Very deep Transformers are typically hard to train because of instability and memory usage, and they are prone to overfitting on a small dataset like Wikitext-103. LayerDrop regularizes the network, reduces the memory usage, and increases training stability as fewer layers are active at each forward pass. These results confirm that this type of approach can be used to efficiently train very deep networks, as shown in Huang et al. (2016) for convolutional networks.

**Sequence to sequence modeling.**   Similarly, as shown in Table 1 and Table 3, applying Layer-Drop to Transformers on text generation tasks such as neural machine translation, summarization, and long form question answering also boosts performance for all tasks. In these experiments, we take the Transformer architectures that are state-the-art and train them with LayerDrop. In neu-

| Model | Enc | Dec | ROUGE-1 | ROUGE-2 | ROUGE-L |
|---|---|---|---|---|---|
| *Abstractive Summarization* | | | | | |
| Transformer (Edunov et al., 2019) | 6 | 6 | 40.1 | 17.6 | 36.8 |
| Transformer + LayerDrop | 6 | 6 | 40.5 | 17.9 | 37.1 |
| Transformer + LayerDrop | 6 | 8 | **41.1** | **18.1** | **37.5** |
| *Long Form Question Answering* | | | | | |
| Transformer Multitask (Fan et al., 2019) | 6 | 6 | 28.9 | 5.4 | 23.1 |
| Transformer Multitask + LayerDrop | 6 | 6 | **29.4** | **5.5** | **23.4** |

Table 3: **Results for CNN-Dailymail Summarization** and **ELI5 QA** (test set).

| Data | Layers | Model | MNLI-m | MRPC | QNLI | SST2 |
|---|---|---|---|---|---|---|
| Books + Wiki | 24 | RoBERTa | 89.0 | 90.2 | 93.9 | 95.3 |
| | 24 | RoBERTa + LayerDrop | **89.2** | 90.2 | **94.2** | **95.4** |
| + more data | 24 | RoBERTa | 90.2 | 90.9 | 94.7 | 96.4 |
| | 24 | RoBERTa + LayerDrop | 90.1 | **91.0** | 94.7 | 96.8 |
| | 48 | RoBERTa + LayerDrop | **90.4** | 90.9 | **94.8** | **96.9** |

Table 4: **Results on Various NLU Tasks** for RoBERTa Large trained for 500K updates (dev set).

ral machine translation on newstest2014, our 12 encoder layer Transformer model with LayerDrop further improves the state of the art, reaching 30.2 BLEU. In comparison, a standard Transformer trained without LayerDrop diverges with 12 encoder layers. This is a known problem, and techniques such as improved initialization could be used to maintain stability (Junczys-Dowmunt, 2019; Zhang et al., 2019; Wang et al., 2019b; Wu et al., 2019b), but are out of the scope of this work. Similar results are seen in summarization.

**Bi-Directional Pre-training.** In a second set of experiments, we look at the impact of LayerDrop on pre-training for sentence representation models and subsequent finetuning on multiple natural language understanding tasks. We compare our models to a variant of BERT for sentence representations, called RoBERTa (Liu et al., 2019), and analyze the results of finetuning for data adaptation on MNLI, MRPC, QNLI, and SST2. We apply LayerDrop during both pre-training and finetuning.

We compare the performance of the large architecture on the BooksCorpus+Wiki dataset used in BERT. We analyze the performance of training on the additional data used in RoBERTa as well as pre-training for even longer. Comparing fixed model size and training data, LayerDrop can improve the performance of RoBERTa on several tasks. LayerDrop can further be used to both enable and stabilize the training (Huang et al., 2016) of models double the size for even stronger performance.

### 5.2 PRUNING TRANSFORMER LAYERS TO ON-DEMAND DEPTH WITH LAYERDROP

**Pruning Generation Tasks.** In Figure 2, we investigate the impact of the number of pruned decoder layers on the performance of a Transformer for language modeling, neural machine translation, and summarization. We compare three different settings: standard Transformer models trained without LayerDrop but subsequently pruned, standard Transformer models trained from scratch to each desired depth, and lastly our approach: pruning layers of a Transformer trained with Layer-Drop. Our model is trained once with the maximum number of layers and then pruned to the desired depth, without any finetuning in the shallower configuration. Our approach outperforms small models trained from scratch, showing that LayerDrop leads to more accurate small models at a whole range of depths. Further, training with LayerDrop does not incur the computational cost of retraining a new model for each desired depth. For completeness, dropping layers of a deep Transformer trained without LayerDrop performs poorly as it was not trained to be robust to missing layers.

**Pruning BERT-like Models.** In Table 7 (left), we compare pruning Transformers trained with LayerDrop to different approaches used to create smaller, shallower models. We compare to BERT

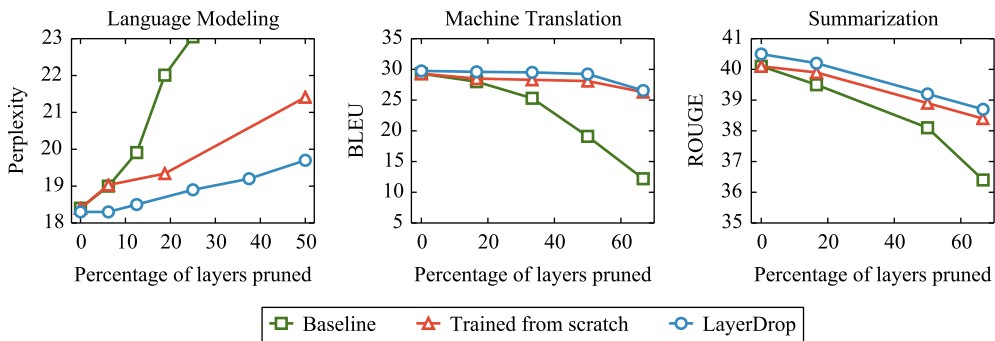

Figure 2: **Performance as a function of Pruning** on various generation tasks (test set), compared to training smaller models from scratch and pruning a Transformer baseline trained without LayerDrop. Pruning networks with LayerDrop performs strongly compared to these alternatives.

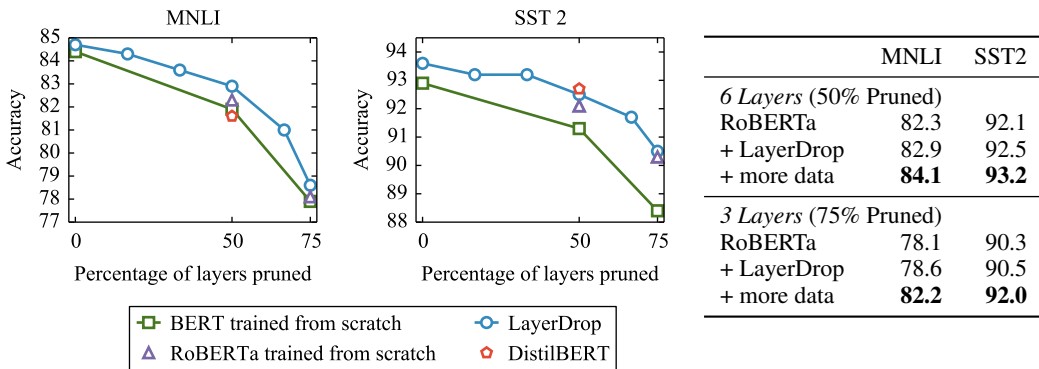

| | MNLI | SST2 |
|---|---|---|
| *6 Layers* (50% Pruned) | | |
| RoBERTa | 82.3 | 92.1 |
| + LayerDrop | 82.9 | 92.5 |
| + more data | **84.1** | **93.2** |
| *3 Layers* (75% Pruned) | | |
| RoBERTa | 78.1 | 90.3 |
| + LayerDrop | 78.6 | 90.5 |
| + more data | **82.2** | **92.0** |

Figure 3: (left) **Performance as a function of Pruning** on MNLI and SST2 compared to BERT and RoBERTa trained from scratch and DistilBERT. Pruning one network trained with LayerDrop (blue) outperforms alternatives that require a new network for each point. (right) **Performance when Training on More Data** shows even stronger results on MNLI and SST2 for pruned models.

base and RoBERTa base trained from scratch with 6 and 3 layers as well as recent work on distillation, called DistilBERT (Sanh, 2019). We analyze both BERT and RoBERTa models as the vocabulary is not the same due to differences in subword tokenization, which affects performance.

DistilBERT occasionally performs worse than BERT of the same size trained from scratch, which confirms the findings of Liu et al. (2018b) about the performance of pruned models compared to training small models from scratch. Our approach, however, obtains results better than BERT and RoBERTa trained from scratch. Further, our method does not need any post-processing: we simply prune every other layer of our RoBERTa model that has been pre-trained with LayerDrop and finetune the small models on each of the downstream tasks, following standard procedure. When training with additional data, shown in Table 7 (right), even stronger performance can be achieved.

## 6    ABLATION STUDIES

**Comparison of Structured Dropout**    Figure 4 (left) contrasts various forms of structured dropout: dropping attention heads, FFN matrices, and entire Transformer layers. Dropping *heads* alone is worse than dropping entire sub-layers or layers. It also offers no advantage in terms of running time as attention heads are computed in parallel for computational efficiency. We observe no large differences between dropping sub-layers and layers, possibly because we are working with relatively shallow networks. In theory, dropping sub-layers should perform better and we expect this to be the

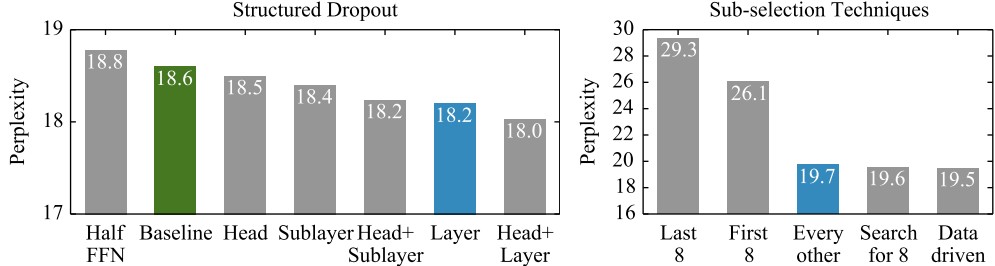

Figure 4: (left) **Impact of Various Structured Dropouts** on Wikitext-103 Valid. Dropping Layers is straightforward and has strong performance. (right) **Comparison of Pruning Strategies** on Wikitext-103 Valid. Marginal gains can be achieved, but dropping every other layer is hard to beat.

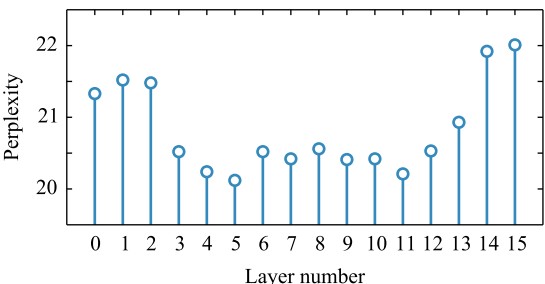

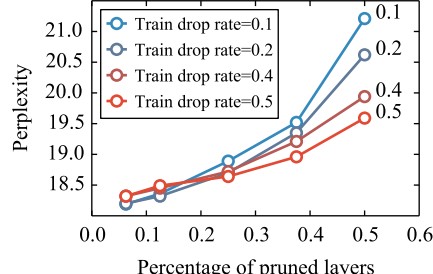

Figure 5: **Relative Importance of Specific Layers.** (Wikitext-103 Valid) The full network is pruned into various 8 layer sub-network configurations, and the average perplexity pruning layer $n$ is displayed above.

Figure 6: **Effect of Train LayerDrop** on Inference-time Pruning. (Wikitext-103 Valid) Training with larger LayerDrop is beneficial for significant pruning.

case with very deep Transformers. We experiment with overlapping structured groups, such as *heads + layers* and *heads + sub-layers* and find that the beneficial effect can be advantageously combined. We focus on layers for simplicity, as dropping more structures introduces more parameters to tune.

**Comparison of Various Pruning Strategies.** Figure 4 (right) contrasts various approaches to sub-selecting model layers at inference time.

The predominant method used in this paper, the straightforward strategy of selecting every other layer, is tough to beat. We find only marginal improvement can be gained by searching over the validation set for the best set of 8 layers to use and by learning which layers to drop. In contrast, dropping chunks of consecutive layers is harmful. Namely, removing the first half or last half of a model is particularly harmful, as the model does not have the ability to process the input or project to the full vocabulary to predict the subsequent word.

**Choosing which Layers to Prune.** Not all layers are equally important. In an experiment on Wikitext-103, we pruned selections of 8 layers at random. Figure 5 displays the perplexity when that layer is removed, averaging results from 20 pruned model per layer. The input and output layers of a network are the most important, as they process the input and project to the output vocabulary.

**Relationship between LayerDrop at Training Time and Pruning at Inference Time.** Figure 6 displays the relationship between the training time LayerDrop and the performance of a pruned network at test time. If significant depth reduction is desired, training with larger LayerDrop is beneficial — this equalizes the train and test time settings. An analysis for BERT is in the Appendix.

## 7 CONCLUSION

Structured dropout regularizes neural networks to be more robust to applying structured pruning at inference time. We focus on the setting where structures are layers, enabling pruning of shallow and efficient models of any desired depth. In a variety of text generation and pre-training tasks, we show that LayerDrop enables and stabilizes the training of substantially deeper networks and simultaneously allows for the extraction of models of various depths with strong performance.

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

# A    APPENDIX

## A.1    ADDITIONAL IMPLEMENTATION DETAILS

### A.1.1    NEURAL MACHINE TRANSLATION

**WMT en-de**: We model a 32K joint byte-pair encoding. We train using the cosine (Loshchilov & Hutter, 2016) learning rate schedule from Wu et al. (2019a) with label smoothing 0.1. vocabulary (Sennrich et al., 2015). We train on 8 GPU for total training time 66k seconds.

**IWSLT de-en**: The dataset consists of 160K training pairs, fully lowercased. We model a 10K joint BPE vocabulary and generate with beam size 4. We do not average checkpoints. Following Wu et al. (2019a), we use the Transformer base architecture with 6 encoder layers and 6 decoder layers. As the dataset is small, we decrease the overall model size and instead use the following parameters: FFN size 1024, hidden dimension 512, and 4 attention heads. We train on 1 GPU.

**Pruning**: We apply the *Every Other Layer* strategy to the decoder and do not finetune.

### A.1.2    LANGUAGE MODELING

**Training**: To handle the large vocabulary of Wikitext-103, we follow Dauphin et al. (2017) and Baevski & Auli (2018) in using adaptive softmax (Grave et al., 2016) and adaptive input for computational efficiency. For both input and output embeddings, we use dimension size 1024 and three adaptive bands: 20K, 40K, and 200K. We use a cosine learning rate schedule (Baevski & Auli, 2018; Loshchilov & Hutter, 2016) and train with Nesterov's accelerated gradient (Sutskever et al., 2013). We set the momentum to 0.99 and renormalize gradients if the norm exceeds 0.1 (Pascanu et al., 2014). During training, we partition the data into blocks of contiguous tokens that ignore document boundaries. At test time, we respect sentence boundaries. We train on 8 GPU for total training time of 216k seconds.

**Pruning**: We apply the *Every Other Layer* strategy and do not finetune.

### A.1.3    SUMMARIZATION

**Data:** We use the full text (non-anonymized) version of CNN-Dailymail introduced by See et al. (2017). Following Fan et al. (2017), we truncate articles to 400 tokens and model a joint byte-pair vocabulary of 32K types (Sennrich et al., 2016).

**Training**: We train using Adam with a cosine learning rate schedule, warming up for 10K steps. We optimize dropout in the range $\{0.2, 0.3\}$ on the validation set and set LayerDrop to 0.2. We train on 1 GPU.

**Pruning**: We apply the *Every Other Layer* strategy to the decoder and do not finetune.

### A.1.4    LONG FORM QUESTION ANSWERING

**Training**: We compare to the full multi-task setting of Fan et al. (2019), where data augmentation and multi-tasking is done at training time to increase the data available. We train on 8 GPU.

**Generation**: We set the minimum length to 150 tokens and the maximum length to 200.

### A.1.5    BI-DIRECTIONAL PRE-TRAINING

**Training**: The base architecture is a 12 layer model with embedding size 768 and FFN size 3072. The large architecture consists of 24 layers with embedding size 1024 and FFN size 4096. For both settings, we follow Liu et al. (2019) in using the subword tokenization scheme from Radford et al. (2019), which uses bytes as subword units. This eliminates unknown tokens. Note this produces a different vocabulary size than BERT (Devlin et al., 2018), meaning models of the same depth do not have the same number of parameters. We train with large batches of size 8192 and maintain this batch size using gradient accumulation. We do not use next sentence prediction (Lample & Conneau, 2019). We optimize with Adam with a polynomial decay learning rate schedule. For

| Hyperparameter | Base | Large |
|----------------|------|-------|
| Number of Layers | 12 | 24 |
| Hidden Size | 768 | 1024 |
| FFN Size | 3072 | 4096 |
| Attention Heads | 12 | 16 |
| LayerDrop | 0.2 | 0.2 |
| Warmup Steps | 24k | 30k |
| Peak Learning Rate | 6e-4 | 4e-4 |
| Batch Size | 8192 | 8192 |

Table 5: Hyperparameters for RoBERTa Pretraining

| Model | BLEU |
|-------|------|
| Transformer (Wu et al., 2019a) | 34.4 |
| Dynamic Conv (Wu et al., 2019a) | 35.2 |
| Transformer + LayerDrop | 34.5 |

Table 6: BLEU for `IWSLT` (test set).

BERT-Base, we use 32 GPU (total training time 171k seconds) and for BERT-Large, we use 128 GPU. For the RoBERTa data setting with more data, we use 512 GPU to train BERT-Large.

**Finetuning**: During finetuning, we hyperparameter search over three learning rate options (1e-5, 2e-5, 3e-5) and batchsize (16 or 32 sentences). The other parameters are set following Liu et al. (2019). We do single task finetuning, meaning we only tune on the data provided for the given natural language understanding task. We do not perform ensembling. When finetuning models trained with LayerDrop, we apply LayerDrop during finetuning time as well.

**Training smaller models**: We train the 6 and 3 layer RoBERTa models following the same settings, but using the smaller number of layers and without LayerDrop. We finetune with the same sweep parameters. The 6 and 3 layer BERT model results are taken from Devlin et al. (2018).

**Training larger models**: We train the 48 layer RoBERTa model with 0.5 LayerDrop so only 24 layers on average are active during a forward pass.

**Pruning**: When pruning RoBERTa models, we use the *Every Other Layer* strategy and finetune without LayerDrop for the smaller models.

## A.2  ADDITIONAL RESULTS

**IWSLT**  Table 6 displays results on the IWSLT de-en dataset. We see small improvement, likely as the network is small and already has a large quantity of regularization with dropout, attention dropout, and weight decay. The Transformer is not the state of the art architecture, and there remains a large gap between the Transformer and the DynamicConv model proposed by Wu et al. (2019a).

**Pruning BERT Models**  The numerical values corresponding to the pruned 6 and 3 layer RoBERTa + LayerDrop models are shown in Table 7.

## A.3  ADDITIONAL ANALYSIS

**Impact of LayerDrop on training time.**  Figure 7 shows the increase in training speed when training with increasingly large quantities of LayerDrop. The words per second were computed on 8 V100 GPUs with 32GB of memory, without floating point 16, for a 16 layer model trained on Wikitext-103. Assuming fixed layer size, LayerDrop removes layers at training time randomly, which increases the training speed almost 2x if dropping half the number of layers.

| Model | Dataset | Layers | MNLI-m | MRPC | QNLI | SST-2 |
|---|---|---|---|---|---|---|
| BERT | Books + Wiki | 6 | 81.9 | 84.8 | - | 91.3 |
| Distil BERT (Sanh, 2019) | Books + Wiki | 6 | 81.6 | 82.4 | 85.5 | 92.7 |
| RoBERTa | Books + Wiki | 6 | 82.3 | 82.5 | 89.7 | 92.1 |
| RoBERTa + LayerDrop | Books + Wiki | 6 | 82.9 | 85.3 | 89.4 | 92.5 |
| RoBERTa + LayerDrop | + more data | 6 | **84.1** | **86.1** | **89.5** | **93.2** |
| BERT | Books + Wiki | 3 | 77.9 | 79.8 | - | 88.4 |
| RoBERTa | Books + Wiki | 3 | 78.1 | 79.4 | 86.2 | 90.3 |
| RoBERTa + LayerDrop | Books + Wiki | 3 | 78.6 | 75.1 | 86.0 | 90.5 |
| RoBERTa + LayerDrop | + more data | 3 | **82.2** | **79.4** | **88.6** | **92.0** |

Table 7: Comparison between BERT base with and without distillation with our RoBERTa base trained with LayerDrop. Our models are pruned before finetuning on each individual task. The numbers from BERT are taken from Devlin et al. (2018).

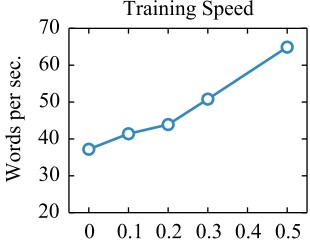

| Model | Valid PPL |
|---|---|
| Pruned w/ LayerDrop | 20.78 |
| + Finetune | 20.56 |

Table 8: **Impact of additional finetuning** on a 16 layer language model pruned to 8 layers.

Figure 7: **Effect of LayerDrop on Training Time**

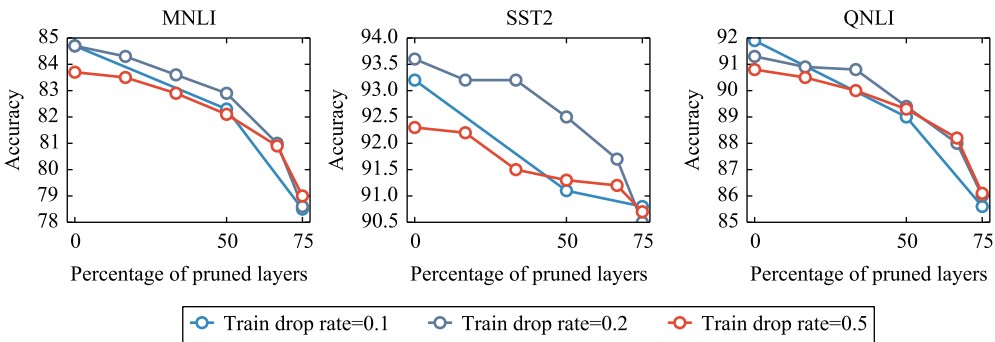

Figure 8: **Effect of Train LayerDrop** on Inference-time Pruning on MNLI, SST2, and QNLI

**BERT: Relationship between LayerDrop at Training Time and Pruning at Inference Time** Similar to the analysis on Language Modeling, we find that training with larger quantities of Layer-Drop allows for more aggressive pruning at inference time on various natural language generation tasks. However, as these tasks involve a finetuning step on the downstream tasks after pre-training, the effect is less straightforward. Results are shown in Figure 8.

**Impact of Finetuning.** LayerDrop allows models to be pruned to the desired depth at test time. Apart from finetuning for data adaptation on the GLUE tasks, we do not finetune the performance of our smaller models on any of the other tasks we consider in this work. As shown in Table 8, we found that finetuning the pruned models only results in marginal improvement. Further, the finetuning parameters were dependent on the depth of the model at test time and difficult to optimize.

| LayerDrop | Dropout | Valid PPL |
|---|---|---|
| 0.5 | 0.1 | 19.03 |
| 0.5 | 0.2 | 19.22 |
| 0.5 | 0.3 | 19.31 |
| 0.5 | 0.4 | 19.62 |
| 0.5 | 0.5 | 19.95 |

Table 9: **Performance Varying Dropout with Fixed LayerDrop** on a 16 layer language model trained on Wikitext-103 (Valid).

| Model | Valid PPL |
|---|---|
| Adaptive Input* | 18.4 |
| Random LayerDrop 0.2 | 18.2 |
| Linear LayerDrop to 0.3 | 18.6 |
| Linear LayerDrop to 0.5 | 18.5 |
| Linear LayerDrop to 0.8 | 18.9 |

Table 10: **Random v. Linear Decay** Layer-Drop on a 16 layer language model trained on Wikitext-103 (Valid). * result is from Baevski & Auli (2018)

| Structured Dropout | Valid PPL |
|---|---|
| Half FFN | 29.6 |
| Baseline | 28.3 |
| Head | 28.1 |
| Sublayer | 19.9 |
| Head + Sublayer | 19.8 |
| Layer | 19.7 |
| Head + Layer | 19.7 |

Table 11: **Performance Varying Structured Dropout and Pruning** to an 8 layer language model trained on Wikitext-103 (Valid). Pruning is done by removing every other layer to half the model size.

**Effect of Varying Standard Dropout.** LayerDrop adds a strong regularization effect to neural network training. We examine the importance of tuning the standard dropout parameter when training with LayerDrop. In Table 9, we show the performance when LayerDrop is fixed and standard Dropout is varied. We see that when training with LayerDrop, the quantity of standard Dropout can be reduced.

**LayerDrop Schedule: Random or Linear.** We investigate the random structured dropping of layers compared to the linear decay schedule proposed in Huang et al. (2016) in Table 10. We find that the linear decay schedule does not provide performance improvement compared to random dropping, which is more straightforward to implement.

**Impact of Types of Structured Dropout when Pruning.** Figure 4 (left) contrasts the performance of various forms of structured dropout, such as dropping attention heads, sub-layers of Transformers such as attention or FFN, portions of FFN matrics, and entire Transformer layers. It examines these results in the setting of evaluating the full depth model on language modeling and shows that in general, different types of structured dropout can improve performance.

In Table 11, we examine the effect of varying training time structured dropout with performance when pruning. We show that the trend shown in Figure 4 is consistent with inference-time pruning performance, particularly that Half FFN dropout performs slightly worse, but other forms of structured dropout are beneficial.

