# OpenReview forum: "Reducing Transformer Depth on Demand with Structured Dropout"
_ICLR.cc/2020/Conference — Accept (Poster)_

### Official Review · AnonReviewer2 · 2019-10-17
**Official Blind Review #2**

**Rating:** 8

**Review:**

This paper presents LayerDrop, a simple method for dropping groups of weights (typically layers) jointly. Despite its simplicity (which is actually a big plus), the method seems to improve performance quite consistently on a range of NLP tasks. Moreover, it allows the authors to train very deep networks, that are very hard to train otherwise (according to the authors). For me the most exciting thing about this approach is that this training regime allows to prune the trained network at test time *without finetuning*, effectively getting a smaller, more efficient network for free. This is a great benefit compared to existing approaches that require retraining a smaller network for each costume size. While the method isn't really applicable to any size, and largely depends on the dropout rate the full model was trained on, I imagine it could serve as a starting point for other researchers to develop more flexible extensions that would allow for any size of network to be pruned at test time. I think this is a very strong submission and strongly advocate accepting it to ICLR.

Questions and comments:

1. The main thing missing for me is some more analysis on the runtime/energetic savings (e.g., in terms of FLOPs) of the proposed method. The authors argue (3.2.1) that approaches such as DropConnect are not necessarily more efficient, but do not analyze the efficiency of their pruned networks apart from the size of the pruned network.

2. Similarly, details about the experiments are also somewhat lacking:
a. how many GPUs were used to train the models? the authors mention 8 v100 in A.3, but I am not sure if this was the setup for all experiments.
b. Figure 7, which shows that LayerDrop also improves training speed, is very interesting and should be part of the main text in my opinion. Was this trend consistent for all experiments?
c. Similarly, presenting the total running time of the models (and not just words per second) would be helpful for reproducibility.
d. Finally, reporting dev and not only test results (e.g., in tables 1 and 2) would also facilitate future reproducibility efforts.

3. Did the authors use a regular dropout? If I understand correctly, in A.1.3, the authors mention tuning the dropout rate between {0.2,0.3}. Was this done for all tasks? and was it done for the baseline models as well? Using dropout in the baseline model with a similar proportion as LayerDrop seems like an important baseline, and in particular it would be interesting to see whether the deep experiments (e.g., 40 layers on WT103) that are hard to train without LayerDrop could converge with regular dropout.

Minor:
- 3.2: "We present *an* regularization approach ..." (should be "a")
- Table 2 is referred to before table 1, it might be clearer to switch them.
- In figure 4, it wasn't clear to me why "Layer" on the lefthand side is much better than "Every other" on the righthand side. Aren't these the same model variant?
- Missing year for paper "Language models are unsupervised multitask learners".


**Experience Assessment:**

I have published one or two papers in this area.

**Review Assessment: Checking Correctness Of Derivations And Theory:**

N/A

**Review Assessment: Checking Correctness Of Experiments:**

I assessed the sensibility of the experiments.

**Review Assessment: Thoroughness In Paper Reading:**

I read the paper thoroughly.

---

> ### Author Response · Authors · 2019-11-13
> **thanks for your review!**
>
> Thanks for your review and all of your comments and questions. We have included our response below, and please let us know if you have additional questions. We have a long response as you had a lot of questions!
>
> Re: runtime (Question 1) - Yes, this is a very important point. Appendix Figure 7 shows that by applying 50% dropout, the model training sees almost 2x speedup (it is not exactly double as the embedding and projection to vocabulary computation does not change). We added inference words per second and total inference time on the validation dataset for Wikitext-103 Language Modeling in the below table. We will update the main paper to include these results as we need to re-format the main paper slightly.
>
> Model Depth    | Inference Speed 1 GPU      | Total Inference Time
> 16 Layers (full) | 1307 tok/sec                         | 166.4 s
> 14 Layers          |  1476 tok/sec                        | 147.4 s
> 12 Layers          |  1684 tok/sec                        | 129.2
> 10 Layers          |  1987 tok/sec                        | 109.5
> 8 Layers (50%) |  2394 tok/sec                        | 90.9
>
> Re: number GPUs (Question 2a) - We have updated Appendix Section A1 to include the number of GPUs used for each experiment in addition to the list of hyperparameters we provide. Specifically, we use 8 GPUs for all tasks except for BERT pretraining as it is very computationally intensive (for summarization and IWSLT de-en, we use 1 GPU). To train BERT Base setting, we use 32 GPUs and to train the BERT Large setting, we use 128 GPUs on the BERT dataset (13 GB of text) and 512 GPUs on the RoBERTa data setting (160 GB of text). We use gradient accumulation to keep the batchsize at the setting of the RoBERTa paper.
>
> Re: consistency of training time improvement (Question 2b) - Yes, as layers are removed from the model randomly during training time on all tasks, those layers are not forwarded through so on all tasks the training speed will improve. You can also see the inference speed improvement in the above table in our response.
>
> Re: Running time (Question 2c) - We have added to Appendix Section A1 the total expected running time of these models. As our experiments are done with LayerDrop = 0.2, we shave about 20% off the total training time. We will also release the pretrained models for all tasks, as it is particularly computationally intensive to train the BERT models at the largest scales.
>
> Re: Report dev results (Question 2d) - Yes, good point, thanks for bringing this up. We have added Table 8 in the Appendix that include dev set results our test set results. Note the ablations are on the dev set already and the BERT comparison is also on dev.
>
> Re: Do we use regular dropout (Question 3) - Yes, on all tasks we use regular dropout. In Section 4, we describe the hyperparameters of regular dropout that we tuned when training our LayerDrop models. For BERT training, we did not tune any parameters as it is very computationally intensive to train one of these models. We do not tune the regular dropout values on the baselines, as we report baselines from the original papers and assumed that the dropout was already tuned to those tasks. We added ablation Table 10 to the Appendix, which shows performance on Language Modeling on Wikitext-103 keeping LayerDrop fixed and tuning standard dropout. Our takeaway is that when LayerDrop is added, the quantity of standard Dropout can be slightly decreased due to the regularization effect of LayerDrop.
>
> Re: Minor comments- thanks! We have made those fixes and appreciate you reading the paper so carefully.
>
> To answer your question about Figure 4- Yes, they are the same model, but the Figure on the right (comparing different types of dropout) is the full model with all of its layers, while the Figure on the left (comparing different strategies to prune) is the same model pruned to 8 layers with the different techniques. The gap in perplexity is from halving the model size. We updated the Appendix to include Table 12 comparing different types of structured dropout when pruning (e.g. the varying setting of the left side but with models pruned with Every Other Layer). Note that some of the pruned results are not competitive because the models have not been trained with LayerDrop and are thus not robust to pruning at inference time (e.g. Half FFN, Baseline, Head Dropout alone).

---

> > ### Comment · AnonReviewer2 · 2019-11-14
> > **Response to your comments**
> >
> > Thank you for the detailed response!
> >
> > Re dev results, it would be nice to see dev results for all experiments, not just the MT ones. Also, figure 3 doesn't mention explicitly whether these are test or dev results.
> >
> > Dropout: I realize you assume dropout was properly tuned for your baselines, but I still think dropping a similar proportion of parameters in your baselines to the ones you drop with LayerDrop is an important baseline, and would be happy to see these results for at least for 1-2 tasks to ensure that the gains are not from standard heavier regularization.
> >
> > Figure 4: I am sorry, but this is still a bit confusing to me (I think you might have mixed left and right in your response). If the structured dropout comparison shows the full model (without LayerDrop?) then what is being compared here?

---

> > > ### Author Response · Authors · 2019-11-15
> > > **Thanks for your additional questions!**
> > >
> > > Thanks for your fast response!
> > >
> > > Re: dev results - We will make clear that the BERT results are already on dev set (and edit the caption of Figure 3) and the LM results as well in the ablations, by adding them to Appendix Table 8 which also shows dev results on WMT and Long Form Question Answering (the first part is on QA and the second on NMT). Thanks for pointing this out - the dev results for Summarization we indeed forgot to add, we will update the paper with this number. We will also format Table 8 to include the specific lines for BERT pre-training and LM to make it very clear so people do not have to cross-reference the main paper when looking at this table.
> > >
> > > Re: Dropout - Yes, you are right. We have Appendix Table 10 to show the performance of keeping LayerDrop fixed to 0.5 and varying Dropout. We will also add the following table, which shows the performance of keeping LayerDrop fixed to 0.2 and varying standard Dropout. This table, in combination with Appendix Table 10 on LayerDrop 0.5, provides a comprehensive analysis on Language Modeling on Wikitext-103 of the performance of standard heavier regularization compared to LayerDrop (in combination with Figure 4, which compares to dropping attention heads, for example). We will update the paper to include an analysis like this for NMT as well, but require a few days to do these experiments and the rebuttal period will be over by then, but we will add it.
> > >
> > > LayerDrop      | Dropout         | Valid PPL
> > > 0.2                    | 0.1                   | 18.5
> > > 0.2                    | 0.2                   | 18.2
> > > 0.2                    | 0.3                   | 18.4
> > > 0.2                    | 0.5                   | 18.7
> > >
> > > We will add a note when describing this table that the standard dropout setting of Baevski et al is 0.3 (described in Section 4.1 of their paper), but a reduction to 0.2 standard Dropout performs better in our setting due to the additional regularization provided by LayerDrop.
> > >
> > >
> > > Re: Figure 4, sorry about the confusion. We meant that Figure 4 (left) shows the performance of a full model trained with structured dropout (with LayerDrop being one of the choices) but with no layers pruned. Specifically, we train the 16 Layer Transformer + LayerDrop with the different structured dropout configurations shown in the bar chart (head dropout, layer dropout, sublayer dropout). There is no inference time pruning, so these results are the perplexities of a 16 layer model.
> > >
> > > On Figure 4 (right), we show the performance of keeping a subset of 8 out of the 16 layers, varying the technique for choosing the 8 layers. This table varies pruning, so only evaluates LayerDrop (while the Left figure does not prune and evaluates different structured dropout). This is why the perplexity on the right figure is around 1.5 PPL worse, as it reflects the loss of 50% of the model capacity due to pruning.
> > >
> > > To address your review point that these two diagrams are not aligned, we added Appendix Table 12 that is trained with Structured Dropout (with LayerDrop being one of the choices), but evaluating perplexity in the pruned to 8 layers regime, where the pruning is fixed at the every other layer technique.
> > >
> > > Let us know if you have any other questions. Thanks for the detailed review!

---

### Official Review · AnonReviewer1 · 2019-10-22
**Official Blind Review #1**

**Rating:** 6

**Review:**

This work explored the effect of LayerDrop training in efficient pruning at inference time. The authors showed that it is possible to have comparable performance from sub-networks of smaller depth selected from one large network without additional finetuning. More encouraging is that the sub-networks are able to perform better than the same network trained from scratch or learned based on distillation.

Besides the promising results, I think the authors could make the presentation more coherent. Since the title is about "reducing transformer depth on demand", the focus is on pruning the network to meet inference requirements. But the authors spent a lot of space showing improved results on many tasks, which are mainly from learning a larger network or with additional data compared to the baselines. Then some of the results shown in the appendix, especially the ones referenced in the main text, could be brought into the main part.

On the other hand, I do not think it is adequate to argue the proposed method is a "novel approach to train over-parameterized networks". As the authors acknowledged, the layer dropping technique has been proposed in (Huang et al., 2016). Even though the authors extended this to different components of the network, the main focus is on layer dropping which is exactly the one proposed in (Huang et al., 2016). Actually, two layer dropping schedules were introduced in (Huang et al., 2016). One is the uniform dropping which is adopted in this work, the other is the linear decay dropping which is shown to achieve better performance (Huang et al., 2016). Even though more involved, it is interesting to see how the linear decay dropping works in terms of pruning.

It is intriguing to see that simple dropping method as every other could perform comparably to exhaustive search as shown in Figure 4 (right). Is this an artifact of the used dropping masks in training or something intrinsic to the method? The Data Driven Pruning approach, in a way, has the same flavor as the recently proposed dynamic inference methods [1,2] reducing the inference on a per-input basis. That is, different inference complexity will be given different inputs based on the inferred difficulty. The proposed method, on the other hand, assigns the same inference complexity to all the inputs but tries to learn strong sub-networks. It is worth mentioning these works and compare the differences.

[1] Z. Wu, T. Nagarajan, A. Kumar, S. Rennie, L.S. Davis, K. Grauman, and R. Feris. BlockDrop: Dynamic inference paths in residual networks. CVPR 2018.
[2] X. Wang, F. Yu, Z.-Y. Dou, T. Darrell, and J.E. Gonzalez. SkipNet: Learning dynamic routing in convolutional networks. ECCV 2018.

**Experience Assessment:**

I have read many papers in this area.

**Review Assessment: Checking Correctness Of Derivations And Theory:**

N/A

**Review Assessment: Checking Correctness Of Experiments:**

I carefully checked the experiments.

**Review Assessment: Thoroughness In Paper Reading:**

I read the paper thoroughly.

---

> ### Author Response · Authors · 2019-11-13
> **thanks for your review**
>
> Thanks for the comments! We appreciate it. Please find our response below and let us know if you have any further questions.
>
> Re: Coherence - Thanks, we agree with you that our focus is pruning, and we tried to present the results in an order that was natural to follow. To address your point that it puts too much emphasis on being able to train deeper models, we could do two things: (1) reduce the size of the section on deeper models by perhaps moving some of the results to the Appendix, or (2) switch the order of the sections such that pruning comes first. Please let us know which you would prefer!
>
> Re: Appendix has lots of results - We wanted to include the results on all of the settings we experimented with in the paper and additional analyses. Note however that Table 7 in the Appendix is the table version of the Figure 3 - we put the numbers in a table so others could compare and see the exact values compared to the Figure.
>
> Re: wording of sentence- Yes, we agree this sentence was not worded well, it was not our intention and thank you for the suggestion to improve the paper. We have adjusted the sentence you pointed out to instead state: “In this work, we propose a novel approach to extract any sub-network without a post-hoc pruning process from over-parameterized networks.” Please let us know what you think.
>
> In general, the core of our work is to show that Structured Dropout can be used for pruning. We focus on the case of layers as it is very natural for pruning to make more shallow models. Our work shows that structured dropout can be generalized beyond layers alone (see Figure 4, left). Further, we show that this technique can be used to create multiple sizes of smaller models from just *one* large model, which means that users would not need to constantly retrain smaller models.
>
> re: Linear Decay Schedule- thanks for pointing this out. Actually we tried this in our experiments, but we found it slightly harmful for model performance. The results are shown in Appendix Table 11.  Unlike Huang et al where there is focus on keeping layers at the top, our experiments show that the middle layers are the least important (see Figure 5). Further, unlike Huang et al, we are not focused on training models with hundreds of layers.
>
> Re: Exhaustive Search - To answer your question about why exhaustive search doesn’t perform that much better, we believe it is because the model is trained in the setting of random dropout, so if you train with 50% LayerDrop, then on average roughly every other layer will remain. Also, exhaustive search has a large potential downside of overfitting to the validation set.
>
> Re: Data Driven Pruning - Yes, we agree. We added citations to these dynamic inference methods.
>
> To answer your question, the main difference is that in our Data Driven Pruning, it is a simpler approach that does not vary the model based on the input layers. Instead, we try to learn based on the dataset which layers are the most relevant, but at inference time forward a *fixed* set of layers, as you describe. We are very excited by the dynamic inference techniques and are interested in exploring them in future work.

---

> > ### Comment · AnonReviewer1 · 2019-11-14
> > **Further comments on the coherence**
> >
> > Thanks for the detailed response.
> >
> > As I mentioned in the review, it is very encouraging to see simple method like layer dropping could provide smaller inference networks beating even the DistilBERT without any additional training.
> >
> > It is good to have some sanity check about the performance of the Transformer network trained with LayerDrop. To my knowledge, I do not think there exists work comparing the interaction between LayerDrop and residual connections. However, the claimed SOTA results almost all come from training with larger network/data. It is hard to know whether LayerDrop is the sole reason responsible for the improvement.

---

> > > ### Author Response · Authors · 2019-11-14
> > > **Thanks for your comment!**
> > >
> > > Thanks for your fast response!
> > >
> > > To address your point about Transformers without LayerDrop:
> > >
> > > (1) In a comparable setting, Transformer + LayerDrop is better than Transformer alone. See Table 1, Row 2 Baseline (29.3) and Row 4 LayerDrop (29.6) for NMT, higher is better. Table 2, Row 1 Baseline (18.7) and Row 3 LayerDrop (18.3) for LM, lower is better. Table 3, Row 1 Baseline (40.1) and Row 2 LayerDrop (40.5) for Summarization, higher is better. Table 4 displays results for BERT style pre-training in two sections: with BERT data only (top half) and with more data (bottom half). With BERT data only, the Baseline has 89.0 on MNLI and LayerDrop has 89.2.
> > >
> > > (2) For the deep SOTA models that we show- deeper models without LayerDrop do not work well because of overfitting and instability during training when the models are deep. For example, on neural machine translation, a 12 layer encoder model does not converge to good results. When we apply other techniques from the literature (see our response to Weng Rongxiang's comment on deeper NMT models), we can achieve BLEU of 28.3 on 12 layer encoder models without LayerDrop. However, our 12 layer model with LayerDrop is much better - we see BLEU 30.2.
> > >
> > > On Language modeling, we see similar trends. There is worse performance with a 24 Layer Transformer than a 16 layer Transformer, due to the depth. By training with LayerDrop as a regularizer, we can improve the performance.

---

### Official Review · AnonReviewer3 · 2019-10-23
**Official Blind Review #3**

**Rating:** 6

**Review:**

The paper proposes a method, LayerDrop, for pruning layers in Transformer based models. The goal is to explore the stochastic depth of transformer models during training in order to do efficient layer pruning at inference time. The key idea is simple and easy to understand: randomly dropping transformer layers during training to make the model robust to subsequent pruning. The authors perform empirical studies on several sequence modeling task to conclude that the proposed approach allows efficient pruning of deeper models into shallow ones without fine-tuning on downstream tasks. There are also empirical experiments done to demonstrate that the proposed approach outperforms recent model pruning techniques such as DistillBERT under comparable configurations.

Strengths:
+ The technique seems to be simple to apply yet powerful and promising.
+ Strong results from the pruned networks without fine-tuning on downstream tasks.
+ Good ablation studies that help establish the connection to other pruning strategies and the internal of LayerDrop.

Weaknesses:
- Stochastic depth has demonstrated a lot of significance for training in prior work. Although the end goal here (for pruning) is slightly different, the novelty is a little incremental.

Overall, the paper is a good contribution given the current great interest of transformer-based models. The motivation is quite clear, and the writing is easy to follow. It is also a sensible approach given the strong regularization effect of stochastic depth.

Question:
Similar to Pham et al.'s work on applying stochastic depth to train very deep transformers for speech, do you expect LayerDrop to be helpful for training very deep transformer-based models for NLP tasks assuming memory is not a big constraint?


**Experience Assessment:**

I have published one or two papers in this area.

**Review Assessment: Checking Correctness Of Derivations And Theory:**

I assessed the sensibility of the derivations and theory.

**Review Assessment: Checking Correctness Of Experiments:**

I carefully checked the experiments.

**Review Assessment: Thoroughness In Paper Reading:**

I read the paper thoroughly.

---

> ### Author Response · Authors · 2019-11-13
> **thanks for your review**
>
> Thanks for the review. We have responded to your points below. Please let us know if you would like to see additional analyses or have further questions.
>
> re: Novelty - We agree that the effects of training speed and regularization have been shown in Huang et al, which we cite in these sections when we display the related results. However, the core of our work is to show that Structured Dropout can be used for pruning. We focus on the case of layers as it is very natural for pruning to make more shallow models. But, our work shows that structured dropout can be generalized beyond layers alone to encompass portions of layers, sublayers such as FFN or attention, and attention heads (see Figure 4, left). Further, we show that this technique can be used to create multiple sizes of smaller models from one large model, which means that users would not need to retrain smaller models of different sizes.
>
> re: Even deeper models - Yes, this is possible. We show in Tables 1, 2, and 4 that this can be used to train models that are double the depth on Language Modeling, Machine Translation, and Sentence Pre-training benchmarks. Applications in ASR are possible as well.  LayerDrop is a strong regularizer and stabilizes training as fewer layers are used each forward pass.
>
> We have added new analyses and results to improve our paper. Appendix Table 10 displays the relationship between LayerDrop and standard Dropout. Appendix Table 12 displays the impact of varying different types of structure dropout (Head, Sublayer, Layer, etc) on pruned networks.

---

### Public Comment · ~Liyuan_Liu2 · 2019-09-30
**Interesting paper and a related work.**

Thanks for your interesting paper. I like the idea to prune w.o. fine-tuning.

We had some similar observations on layer-wise dropout and pruning language models without fine-tuning [1]. Specifically, we replace the residual connection with the dense connection, which allows us to drop any layers without deleting all subsequent ones. Although our method requires some modifications before being applied to BERT (as it requires the dense connection), I think these two methods are very related and have similar intuitions.

Besides, in our experiments, we have an interesting observation: we found the shape of the final network (after pruning) have some randomness. We conjecture this is because the network trained with the layer-wise dropout, is actually an ensemble of many small networks (similar to the lottery ticket hypothesis [2]), and the pruning is actually trying to select one from these networks.

1. Liu, Liyuan, et al. "Efficient Contextualized Representation: Language Model Pruning for Sequence Labeling." EMNLP 2018.
2. Frankle, Jonathan, and Michael Carbin. "The lottery ticket hypothesis: Finding sparse, trainable neural networks." ICLR 2019.

---

> ### Author Response · Authors · 2019-10-01
> **Response to your comment**
>
> Thanks for your comment and sharing your related work! We will add a citation describing your previous paper on dense connections for LSTMs.
>
> Re: ensemble of smaller networks: We agree. The original Dropout paper had a nice interpretation as bagging several smaller models at training time, as you describe. At inference time, we find our method robust to the choice of which layers are pruned, possibly a result of this.

---

### Public Comment · ~Weng_Rongxiang1 · 2019-10-11
**Comments about NMT experiment**

In Table 1, I think authors should compare with the base model with 12 layers encoder.  The 12/6 layers Transformer can be trained easily with a minor modification (post-norm to pre-norm) [1].  Furthermore, I am also interested in the comparison of advanced deep NMT models [1,2,3] with the same setting, e.g. 20 layers encoder. The proposed layer dropout may work with them to improve the effect and efficiency.



[1] Wang et al.,  Learning Deep Transformer Models for Machine Translation. ACL 2019.
[2] Wu et al., Depth Growing for Neural Machine Translation. ACL 2019.
[3] Zhang et al., Improving Deep Transformer with Depth-Scaled Initialization and Merged Attention. EMNLP 2019.

---

> ### Author Response · Authors · 2019-10-17
> **Response to your comment**
>
> Thanks for your comment and for pointing out these references. We will add citations when we update the paper draft. When we use the pre-norm modification you suggest (from reference [1]) to train a Transformer Big architecture with 12 encoder layers, the model converges but does not achieve strong BLEU (we see 28.3).
>
> For the comparison to the additional works you cited: We agree that our proposed techniques can be combined with these other works for improved results. However, we believe adding our techniques to a large variety of existing models is out of scope for this work. Adding LayerDrop allows for the training of 20 layer encoders, but we find the other model parameters need to be tuned to prevent overfitting on WMT en-de.

---

### Author Response · Authors · 2019-11-13
**Additional result incorporating Layer Sharing**

We added an additional experiment, an investigation of the question Can LayerDrop be combined with Layer Sharing?

We will add the following table that shows that layers can be shared and LayerDrop can be applied to them. As layer sharing reduces the parameter size, performance is expected to decrease as more layers are shared. However, when sharing chunks of two layers (e.g. layer 0 and layer 1 have the same weights, layer 2 and layer 3 have the same weights, etc) we only see a marginal effect on performance but about 50% fewer parameters. Extending to sharing larger quantities of layers, such as chunks of four layers, we see small decreases in performance likely as the model capacity as been reduced by the loss of parameters from layer sharing. Additional ways of adding back the model capacity could be examined in future work.

Model                                           | Valid
Adaptive Inputs                          | 18.4
Adaptive Inputs + LayerDrop   | 18.2
LayerDrop Share Chunks of 2  | 18.2
LayerDrop Share Chunks of 4  | 18.9

We will add these results into the appendix.

---

### Decision · Program_Chairs · 2019-12-19

**Decision:**

Accept (Poster)

**Comment:**

This paper presents Layerdrop, which is a method for structured dropout which allows you to train one model, and then prune to a desired depth at test time. This is a simple method which is exciting because you can get a smaller, more efficient model at test time for free, as it does not need fine tuning. They show strong results on machine translation, language modelling and a couple of other NLP benchmarks. The reviews are consistently positive, with significant author and reviewer discussion. This is clearly an approach which merits attention, and should be included in ICLR.